# Not So Similar: Different Ways of Nb(V) and Ta(V) Catecholate Complexation

**DOI:** 10.3390/molecules28134912

**Published:** 2023-06-22

**Authors:** Pavel A. Abramov, Maxim N. Sokolov

**Affiliations:** 1Nikolaev Institute of Inorganic Chemistry SB RAS, 3 Akad. Lavrentiev Ave., 630090 Novosibirsk, Russia; 2Research School of Chemistry and Applied Biomedical Sciences, Tomsk Polytechnic University, 634034 Tomsk, Russia

**Keywords:** catechol, niobium, tantalum, crystal structures

## Abstract

The reactions between catechol (H_2_cat) and niobium(V) or tantalum(V) precursors in basic aqueous solutions lead to the formation of catecholate complexes of different natures. The following complexes were isolated and characterized by single-crystal X-ray diffraction (SCXRD): (**1**) (NH_4_)_3_[NbO(cat)_3_]∙4H_2_O; (**2**) K_2_[Nb(cat)_3_(Hcat)]·2H_2_cat·2H_2_O; (**3**) Cs_3_[NbO(cat)_3_]·H_2_O; (**4**) (NH_4_)_4_[Ta_2_O(cat)_6_]·3H_2_O; (**5**) Cs_2_[Ta(cat)_3_(Hcat)]·H_2_cat; (**6**) Cs_4_[Ta_2_O(cat)_6_]·7H_2_O. The isolated crystalline products were characterized by elemental analysis, X-ray powder diffraction (XRPD), FTIR, and TGA. The structural features of these complexes, such as {Ta_2_O} unit geometry, Cs-π interactions, and crystal packing effects, are discussed.

## 1. Introduction

The chemistry of Nb and Ta in oxidation state 5+ is typically associated with polyoxometalates (POM), which have been extensively studied in the last decade by the research groups of Nyman [1,2,3,4], Niu [5,6,7,8], and Sokolov [9,10,11,12,13]. Compared to POM chemistry, there are two relatively small branches of group 5 chemistry dealing with non-organometallic mononuclear M(V) complexes: (i) peroxocomplexes and (ii) [MOL_3_]^n−^ coordination compounds with selected organic ligands. These complexes are of potential interest as catalysts for organic oxidation reactions [14,15,16,17,18]. The [MOL_3_]^n−^ are mostly represented by oxalate complexes studied by Planinic, Juric et al. [19,20,21,22]. Recently, we have added novel photoactive salts into this family [23] and reported a binuclear oxalate complex [Nb_2_O_2_(C_2_O_4_)_4_(µ-C_2_O_4_)]^4−^ [24]. Outside the oxalate field, Loiseau et al. reported interesting examples of polynuclear niobium(V)-based carboxylate oxo complexes obtained by controlled hydrolysis of [Nb(OEt)_5_] in the presence of different carboxylic acids [25,26]. Among the few non-carboxylate complexes, there are two structurally characterized complexes of M(V), which should be taken into consideration. The oxo-tris-(tropolonato)niobium(V) monohydrate was prepared by Drew et al. [27] by hydrolysis of tetrakis-(tropolonato)niobium(V) complex [28]. The tris(1,2-dimethyl-3-hydroxy-4(1*H*)-pyridone)oxotantalum(V) was reported by Mazzi et al. [29] as a new agent for ^178^Ta radiopharmaceutical applications.

Catecholates belong to redox-active ligands, which can be reversibly oxidized in two consecutive steps and can act as electron reservoirs for activation for both substrates and metal centers. They can also be used to design novel magnetic and optic materials [30,31,32]. Regarding the catecholates, it was apparently Rosenheim who in 1932 reported the dissolution of niobic acid in alkaline catechol (C_6_H_4_(OH)_2_, H_2_cat) solutions [33]. Later, in 1959, Brown and Land studied this reaction with UV-VIS techniques [34,35]. This approach was used in the analytical chemistry of Nb and Ta [36]. Several salts of [NbO(cat)_3_]^3−^ and [Ta_2_O(cat)_6_]^4−^ were isolated but not structurally characterized [37].

The redox behavior of Nb(V) and Ta(V) complexes with catecholate derivatives under air-free conditions is quite interesting but rare [38,39]. The most straightforward results in the field belong to Ta(V) complexes with *N*,*N*-bis(3,5-di-*tert*-butyl-2-phenoxide)amide, which is active in the four-electron oxidative formation of aryl diazenes [40].

Interestingly, Nb and Ta, which are commonly regarded as “chemical twins”, give products with different stoichiometry. In this work, we took up these studies in order to clarify the nature of the catecholate complexes formed by Nb(V) and Ta(V) in aqueous solutions. 

## 2. Results and Discussion

### 2.1. Niobium Complexes

Hydrated Nb_2_O_5_·xH_2_O or hexametalate alkali-metals salts A_8_[M_6_O_19_]·nH_2_O were used for the reaction with catechol (C_6_H_4_O_2_, H_2_cat) under air-free conditions. We used different bases (ammonia or different alkalis) to adjust the reaction medium. All synthetic details are summarized in the Experimental section.

Nb_2_O_5_·xH_2_O slightly dissolves in the solution of catechol in aqueous ammonia upon refluxing in an argon atmosphere. As the reaction proceeds, the solution turns from colorless to red. Slow cooling of the reaction mixture gives yellowish crystals of (NH_4_)_3_[NbO(cat)_3_]∙4H_2_O (**1**). The product was isolated by suction filtration and characterized with single-crystal X-ray diffraction (SCXRD). The crystal structure of [NbO(cat)_3_]^3−^ is shown in Figure 1.

The Nb(V) in the structure of [NbO(cat)_3_]^3−^ has a pentagonal-bipyramidal arrangement with short Nb=O and long Nb-O(cat) axial bonds: d(Nb1–O7) = 1.764(5); d(Nb1–O6) = 2.148(5) Å, respectively. The equatorial Nb-O bond distances fall between 2.066(5) and 2.131(5) Å. Such a coordination environment is typical for known mononuclear complexes of Nb^V^ with NbO^3+^ groups [19,20,21,23,41,42,43,44]. It is worth mentioning that in polyoxoniobates, Nb, as a rule, prefers octahedral arrangements with one short and one long axial Nb-O bond. The generation of polyoxoanions containing Nb with CN 7 typically needs hydrothermal treatment [13,45]. In the crystal structure of **1,** we cannot discriminate between NH_4_^+^ and H_2_O positions due to the quality of the SCXRD data (poorly diffracted crystals). 

Switching to KOH as a reaction medium results in the crystallization of K_2_[Nb(cat)_3_(Hcat)]·2H_2_cat·2H_2_O (**2**) by slow cooling of the reaction solution. The structure of [Nb(cat)_3_(Hcat)]^2−^ is shown in Figure 2, left.

In this study, complexes with Nb^V^ coordination environments have not been reported. In this complex anion, Nb has CN 7 with a pentagonal-bipyramidal coordination polyhedral, but without any Nb=O bond. Instead, this position is occupied by a phenolic oxygen from a monodentately coordinated Hcat^−^ anion. The Nb-O axial bond distances are 1.9623(9) Å (Hcat) and 2.0033(9) Å (cat). The equatorial Nb-O bond distances fall into a 2.0033(9)–2.1211(10) Å interval. Moreover, the anion structure is stabilized by the intramolecular H-bond, d(H18-O6) = 1.922(2) Å. This structure can be regarded as a kind of intermediate between [NbO(cat)_3_]^3−^ and [Nb(cat)_4_]^3−^. It should be noted that only a single complex of Nb with 4 O-donor bidentate ligands, and this is of Nb^IV^, has been structurally characterized [46]. In the crystal structure, K^+^, H_2_O, [Nb(cat)_3_(Hcat)]^2−^, and neutral H_2_cat molecules combine into linear neutral associates (Figure 2, right).

The use of Cs_8_[Nb_6_O_19_]·14H_2_O and 2M CsOH under the same reaction conditions leads to the crystallization of Cs_3_[NbO(cat)_3_]·H_2_O (**3**). The geometry of this Nb complex is practically the same as **1**. The most interesting feature of this structure is the location of Cs^+^ cations. Curiously, crystallization from the aqueous solution does not supply water molecules to the cesium coordination sphere. Instead, Cs^+^ prefers Cs^+^-π interactions with the aromatic catechol rings of adjacent [NbO(cat)_3_]^3−^ complexes (Figure 3). A search of the Cambridge Structural Database [47] yielded few other examples of such coordination, but in all cases, Cs^+^ was involved in both Cs-π and Cs-OH_2_ bonding [48,49].

### 2.2. Tantalum Complexes

In the case of Ta, we used only hexatantalates in the reactions with catechol. Using K_8_[Ta_6_O_19_]·16H_2_O as a tantalum source and ammonia as a base resulted in the formation of (NH_4_)_4_[Ta_2_O(cat)_6_]·3H_2_O (**4**), from which single crystals were isolated and characterized by SCXRD. In the crystal structure, a slightly bent {Ta-O-Ta}^8+^ binuclear unit is coordinated with six cat^2−^ ligands (Figure 4). The orientational disorder of the {Ta(cat)_3_} moiety over two close positions (0.5/0.5 occupancies) gives two closely similar geometries of the [Ta_2_O(cat)_6_]^4−^ complexes in the structure: (i) more linear but less symmetrical, with the following parameters: Ta1-O9-Ta2B = 156.7° and d(Ta1-O9) = 1.881(5) Å, d(Ta2B-O9) = 1.950(7) Å; (ii) more symmetrical, with Ta1-O9-Ta2A = 150.8° and d(Ta2A-O9) = 1.928(7) Å. 

The reaction of Cs_8_[Ta_6_O_19_]·14H_2_O (0.37 mmol) with H_2_cat (13.3 mmol) in the presence of ammonia gives Cs_2_[Ta(cat)_3_(Hcat)]·(H_2_cat) (**5**) isolated as a crystalline material after the reaction mixture cools. The geometry of [Ta(cat)_3_(Hcat)]^2−^ (Figure 5, left) is the same as that of the Nb analog **2**. The bidentate Ta-O(cat) distances vary within a 1.993(2)–2.117(2) Å range. The Ta-O(Hcat) is the shortest at 1.936(2) Å. It should be noted that the opposite Ta-O bond distance is 1.993(2) Å, indicating a slight axial compression of the {TaO_7_} bipyramid. Cs^+^-π interactions in the crystal structure of **5** is shown in Figure 5, right.

Decreasing the catechol load to 6.63 mmol with the same load of tantalum precursor leads to the formation of a Cs_4_[Ta_2_O(cat)_6_]·7H_2_O (**6**) crystalline product. In the crystal structure, there are two symmetrically independent [Ta_2_O(cat)_6_]^4−^ units, which differ in their Ta-O bond lengths in the {Ta_2_O} group. The less symmetric dimer of the first type (**A**) has the following distances in {Ta_2_O units}: d(Ta1-O7) = 1.899(6) Å and d(Ta2-O7) = 1.927(6) Å. In the second type of dimer (**B**), both Ta-O (1.9091(4) Å) distances are equal.

The comparison of the [Ta_2_O(cat)_6_]^4−^ ligand’s orientation in **4** and **6**, shown in Figure 6, indicates a possible rotation of one of the {Ta(cat)_3_} units. It appears that [Ta_2_O(cat)_6_]^4−^demonstrates stereochemical flexibility regarding (i) the degree of {Ta_2_O} linearity and equality/inequality of the Ta-O bond distances; and (ii) the rotation of the {Ta(cat)_3_} unit around the Ta-O-Ta core. For comparison, in [Fe(phen)_3_][Ta_2_OF_10_], the Ta–O–Ta angle is 177.2° and the two Ta–O distances are 1.895 and 1.896 Å [50]. Practically the same geometry of [Ta_2_OF_10_]^2−^ (d(Ta-O) = 1.875(1) Å, angle Ta-O-Ta = 180°) has been reported for (NEt_4_)_2_[Ta_2_OF_10_] [51]. The change of F for Cl does not affect the anion geometry for (PPh_4_)_2_[Ta_2_OCl_10_]·2CH_3_CN: d(Ta-O) = 1.875(1) Å; angle Ta-O-Ta = 180° [52]. By contrast, in the structure of peroxolactate (NH_4_)_4_[Ta_2_(C_3_H_4_O_3_)_4_(O_2_)_2_O]·3H_2_O (both Ta-O bonds are equal (1.919(76) Å); angle Ta-O-Ta 163.99(6)°), the presence of bulky lactate ligands affects the linearity of the Ta-O-Ta unit [53].

In 2005, Holland et al. reported a synthesis of [M(cat)_2_(Hcat)(py)] (M = Nb, Ta) complexes by reacting [M(OEt)_5_] with catechol in pyridine under air-free conditions [54]. Careful avoidance of water and O_2_ precludes the appearance of the {M=O} group, and, in this case, Nb and Ta behave in identical ways. 

On the other hand, the reluctance of Ta to form {TaO}^3+^ upon complexation of catecholate is in line with well-known differences in the chemistry of Nb and Ta: in HCl, they form [NbOCl_5_]^2−^ and [TaCl_6_]^−^, respectively; TaOCl_3_ is much less stable than NbOCl_3_; in 2–3% HF solutions, Nb is present as [NbOF_5_]^2−^, while Ta forms [TaF_7_]^2−^. Quantum chemical calculations show that in [MOCl_5_]^2−^ (M = Nb, Ta, Pa, Db), the Ta=O bond is the weakest. This may be due to relativistic effects, which strongly stabilize the d-level in Ta and thus make it less accessible for dative π-bonding [55].

Reported synthetic and structural data for Nb(V) and Ta(V) catecholate complexes can thus be presented in the following scheme (Figure 7). 

### 2.3. Comments on IR-Spectra and Stability

One highly effective way to check the redox state of redox-active catecholates is to consult IR spectroscopy data [56]. According to the collected data in the literature, the presence of ring breathing at 1480 cm^−1^ and C-O stretches between 1250–1275 cm^−1^ can be attributed to catecholate forms. In the spectra of the presented complexes, the following stretches can be found: 1481 cm^−1^, 1251 cm^−1^ (**1**); 1479 cm^−1^, 1253 cm^−1^ (**2**); 1479 cm^−1^, 1253 cm^−1^ (**3**); 1482 cm^−1^, 1249 cm^−1^ (**4**); 1478 cm^−1^, 1257 cm^−1^ (**5**); 1479 cm^−1^, 1248 cm^−1^ (**6**).

None of the reported complexes can be recrystallized from aqueous solutions either in air or in air-free conditions. After dissolution in water in air, the obtained solution immediately changes color from yellow to dark brown (practically black). The ^1^H and ^13^C NMR spectra of such solutions cannot be adequately assigned due to the presence of a huge number of species. The same type of speciation was found after dissolution of the corresponding solid samples in water under air-free conditions.

The solid samples of isolated compounds change color from yellow to dark brown in an air atmosphere. In argon, the color change does not proceed. The phase purity check by XRPD using initial and aged samples does not reflect any significant difference, indicating changes only to the surface of the crystals.

## 3. Materials and Methods

### 3.1. General Information

The catechol and ammonia solutions were of commercial quality (Sigma Aldrich, St. Louis, MO, USA) and were used as purchased. The K_7_[HNb_6_O_16_]·13H_2_O and Cs_8_[Nb_6_O_19_]·14H_2_O were synthesized using the same data as previously reported by Abramov et al. [57]. All syntheses described below were carried out in an Ar atmosphere using the Schlenk technique. The Nb_2_O_5_·xH_2_O was prepared by acidifying K_7_[HNb_6_O_16_]·13H_2_O aqueous solution and was used as freshly prepared.

An elemental analysis was carried out on a Eurovector EA 3000 CHN analyzer (Pavia, Italy). The FT-IR spectra were recorded on an FT-801 spectrometer (Simex, Novosibirsk, Russia). The TGA experiments were conducted using a NETZSCH TG 209 F1 device in an Al crucible by heating a sample from 20 to 300 °C with a 10 °C gradient. The TGA data are summarized in Appendix A.

### 3.2. Synthesis


**Synthesis of (NH_4_)_3_[NbO(cat)_3_]∙4H_2_O (1)**
**:**
A total of 2 g of catechol (18.6 mmol) was added to a suspension of 1 g Nb_2_O_5_·xH_2_O (3.1 mmol) in 20 mL of H_2_O. After the addition of 4 mL of NH_3_·H_2_O, the resulting reaction mixture was refluxed for 1.5 h until the clear-red solution formed. Small yellowish crystals formed after the solution was kept at 5 °C for 3 days. The crude product was collected by filtration on a glass filter, washed with diethyl ether, and dried in vacuo. The typical yield was 0.740 g (23% based on Nb_2_O_5_). Anal. Calc. for C_18_H_26_N_3_NbO_9_ C, H, N (%): 41.5, 5.0, 8.1. Found C, H, N (%): 41.2, 5.2, 8.0.IR (ATR, ν, cm^−1^): 1570(w), 1481(m), 1444(w), 1391(w), 1328(w), 1251(s), 1097(m), 1020(m), 899(w), 866(w), 830(w), 806(s), 734(s), 598(vs).TGA: weight loss 14%—4.4H_2_O.
**Synthesis of K_2_[Nb(cat)_3_(Hcat)]·2H_2_cat·H_2_O**
**(2)**
**:**
A total of 2 g of catechol (18.6 mmol) was added to a suspension of 1 g Nb_2_O_5_·xH_2_O (3.1 mmol) in 20 mL of H_2_O. After adding 4 mL of 1M KOH, the resulting mixture was refluxed for 1.5 h until a clear-brown solution formed. Small yellowish crystals formed after the solution was cooled to 5 °C and kept in such conditions for 7 days. The crude product was collected by filtration on a glass filter, washed with diethyl ether, and dried in vacuo. Yield—several crystals (manual isolation). The XRPD from the bulk sample was hard to assign.IR (ATR, ν, cm^−1^): 3522(w), 1601(w), 1512(w), 1479(s), 1452(w), 1384(w), 1356(w), 1331(w), 1280(w), 1253(s), 1208(w), 1180(w), 1165(w), 1096(m), 1034(w), 1025(w), 1017(w), 896(w), 868(w), 851(w), 805(m), 737(vs), 628(w), 615(w).TGA: weight loss 2.5%—1H_2_O.
**Synthesis of Cs_3_[NbO(cat)_3_]·H_2_O (3):**
A total of 2.13 g of catechol (19.3 mmol) was added to a clear solution of 2.34 g Cs_8_[Nb_6_O_19_]·14H_2_O (1.1 mmol) in 20 mL of H_2_O. After adding 5 mL of 2M CsOH, the resulting mixture was refluxed for 2 h until a clear-brownish solution formed. Red crystals formed after the solution was kept at 5 °C for 7 days. The crude product was collected by filtration on a glass filter, washed with diethyl ether, and dried in vacuo. Yield—several crystals (manual isolation). The XRPD from the bulk sample was hard to assign.IR (ATR, ν, cm^−1^): 3522(w), 1601(w), 1512(w), 1479(s), 1453(w), 1384(w), 1356(w), 1331(w), 1280(w), 1253(s), 1208(w), 1180(w), 1165(w), 1096(m), 1034(w), 1017(w), 908(w), 896(w), 868(w), 851(w), 805(m), 769(w), 737(vs), 628(w), 615(w), 582(w), 564(w).TGA: weight loss 2.5%—1H_2_O.
**Synthesis of (NH_4_)_4_[Ta_2_O(cat)_6_]·3H_2_O (4):**
A total of 1 g of catechol (9.1 mmol) was added to a solution of 1 g K_8_[Ta_6_O_19_]·16H_2_O (0.5 mmol) in 20 mL of H_2_O. After adding 5 mL of NH_3_·H_2_O, the resulting mixture was refluxed for 2 h. After that, the hot yellow solution was filtered from the white precipitate, and 50 mL of EtOH was added to the solution. Crystals formed after the solution was kept below 5 °C for one week. The crude product was collected by filtration on a glass filter, washed with diethyl ether, and dried in vacuo. The typical yield was 0.485 g (26% based on hexatantalate). Anal. Calc. for C_36_H_46_N_4_O_16_Ta_2_ C, H, N (%): 37.5, 4.0, 4.9. Found C, H, N (%): 37.5, 4.2, 4.7.IR (ATR, ν, cm^−1^): 1581(w), 1482(s), 1449(w), 1423(w), 1413(w), 1343(w), 1335(w), 1249(vs), 1150(w), 1099(w), 1044(w), 1019(w), 909(w), 882(w), 869(w), 841(w), 808(m), 769(w), 739(m), 653(w), 610(vs).TGA: weight loss 4%—2.6H_2_O.
**Synthesis of Cs_2_[Ta(cat)_3_(Hcat)]·H_2_cat (5)**
**:**
A total of 1.46 g of catechol (13.3 mmol) was added to a solution of 1 g Cs_8_[Ta_6_O_19_]·14H_2_O (0.37 mmol) in 20 mL of H_2_O. After adding 5 mL of NH_3_·H_2_O, the resulting mixture was refluxed for 2 h. After that, the hot yellow solution was filtered from the white precipitate. Crystals formed after the solution was cooled down to room temperature. The crude product was collected by filtration on a glass filter, washed with diethyl ether, and dried in vacuo. The typical yield was 0.643 g (26% based on hexatantalate). Anal. Calc. for C_30_H_23_Cs_2_O_10_Ta C, H (%): 36.4, 2.3. Found C, H (%): 36.6, 2.6.IR (ATR, ν, cm^−1^): 1478(vs), 1448(w), 1257(w), 1246(w), 1223(w), 1207(w), 1186(w), 1151(w), 1100(m), 1039(w), 1024(w), 953(w), 903(m), 879(w), 871(w), 858(w), 840(w), 821(w), 804(w), 795(w), 758(w), 739(s), 610(m), 582(w), 567(w).
**Synthesis of Cs_4_[Ta_2_O(C_6_H_4_O_2_)_6_]·7H_2_O (6)**
**:**
A total of 0.73 g of catechol (6.63 mmol) was added to a solution of 1.0 g Cs_8_[Ta_6_O_19_]·14H_2_O (0.37 mmol) in 20 mL of H_2_O. After adding 5 mL of NH_3_·H_2_O, the resulting mixture was refluxed for 2 h. After that, the hot yellow solution was filtered from the white precipitate, and 50 mL of EtOH was added to the solution. Crystals formed after the solution was kept below 5 °C for one week. The crude product was collected by filtration on a glass filter, washed with diethyl ether, and dried in vacuo. The typical yield was 0.485 g (26% based on hexatantalate). Anal. Calc. for C_36_H_38_Cs_4_O_20_Ta_2_ C, H (%): 25.7, 2.3. Found C, H(%): 25.3, 2.2.IR (ATR, ν, cm^−1^): 1581(w), 1549(w), 1530(w), 1513(w), 1479(s), 1450(w), 1431(w), 1413(w), 1344(w), 1334(w), 1274(w), 1248(m), 1221(w), 1153(w), 904(w), 868(w), 840(w), 806(m), 767(w), 732(m).TGA: weight loss 7%—6.5H_2_O.

### 3.3. X-ray Powder Diffraction

A powder X-ray diffraction analysis of polycrystals was performed using a Shimadzu XRD-7000 diffractometer (Kyoto, Japan) (CuK-alpha radiation, Ni-filter, linear One Sight detector, 5–50° 2θ range, 0.0143° 2θ step, 2 s per step). The collected data are present in Appendix A.

### 3.4. X-ray Diffraction on Single Crystals

The crystallographic data and refinement details are given in Appendix A. Structures were solved by SHELXT [58] and refined by a full-matrix least-squares treatment against |F|^2^ in anisotropic approximation with SHELXL 2019/3 [59] in the ShelXle program [60]. The main bond distances are given in Appendix A. The crystal packing projections are given in Appendix A. The ellipsoid plots for all complexes are collected in Appendix A.

## 4. Conclusions

This research reports preparation protocols for Nb^V^ and Ta^V^ catecholate complexes in aqueous solutions. The reaction conditions presented provide opportunities to use hydrated metal oxides or hexametalates as starting materials. The direct reaction with catechol is a new point in the chemistry of polyoxoniobates and tantalates, generating many possibilities for future research. The choice of a reagent for basic media generation is very important and affects the formation of the final product. The load of catechol is also very important and allows one to control the formation of {M(cat)_3_} units: the excess generates [M(cat)_3_(Hcat)]^2−^ (M = Nb, Ta) complexes (as 1:4 form), while its absence results in the formation of [NbO(cat)_3_]^3−^ or [Ta_2_O(cat)_6_]^4−^ (as 1:3 form). Consequently, we propose that an equilibrium is established between 1:4 and 1:3 forms in the reaction solution. In complexation with catecholate, Nb and Ta follow their usual pattern: while Nb forms {NbO}^3+^, Ta avoids the formation of {TaO}^3+^, preferring instead to form the slightly bent {Ta_2_O}^8+^ to minimize Ta-O π-bonding. 

In order to obtain more extensive information about intermediate or more complex species, additional studies are needed. Another challenging point is catching the species generated during the solvation of the reported complexes. At the current stage, NMR data show the formation of plenty of complexes after the dissolution of the titled compounds in water or DMSO.

## Figures and Tables

**Figure 1 molecules-28-04912-f001:**
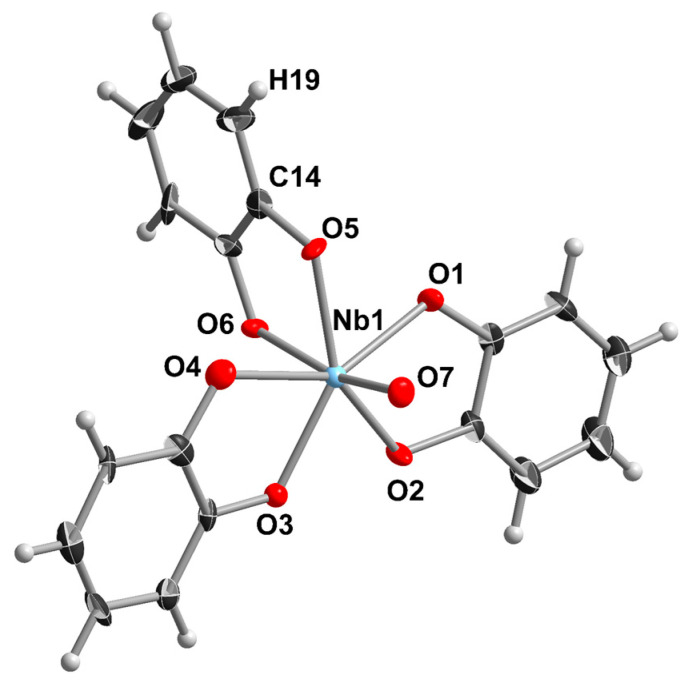
The structure of [NbO(cat)_3_]^3−^ (ellipsoid plot with 50% probability) in the crystal structure of **1**.

**Figure 2 molecules-28-04912-f002:**
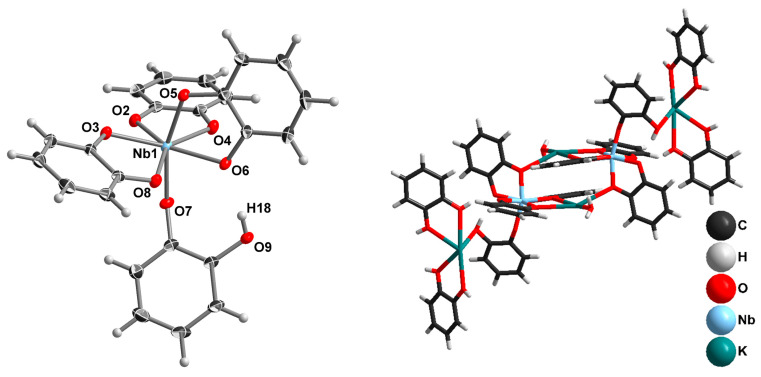
The structure of [NbO(cat)_3_(Hcat)]^2−^ in ellipsoids with 50% probability (**left**). The structure of {K_4_(H_2_O)_2_(H_2_cat)_4_[NbO(cat)_3_(Hcat)]_2_} associate (**right**).

**Figure 3 molecules-28-04912-f003:**
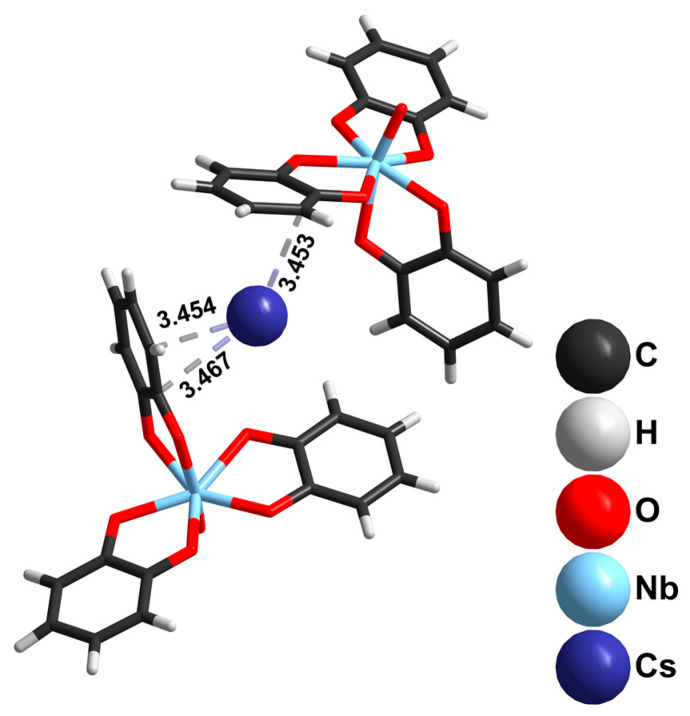
The location of the Cs^+^ cation in the crystal structure of **3**.

**Figure 4 molecules-28-04912-f004:**
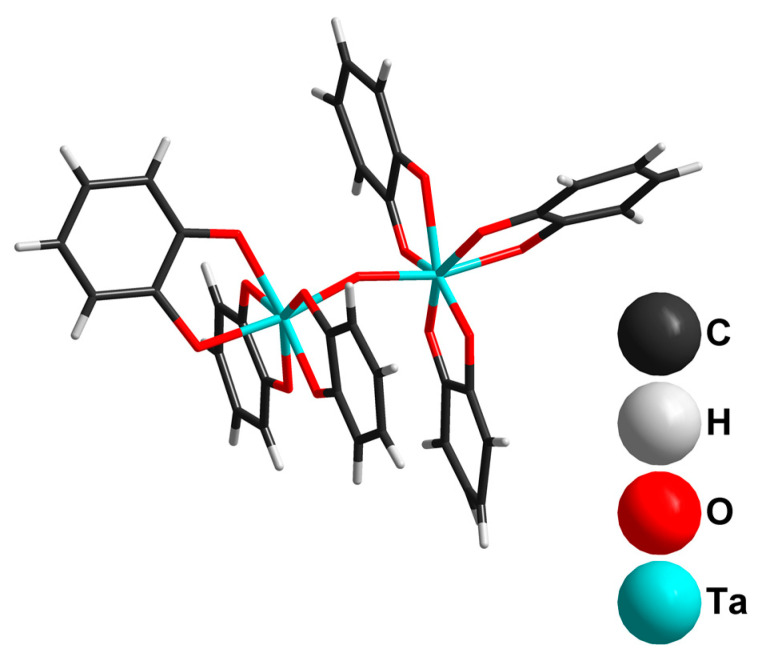
The structure of the [Ta_2_O(cat)_6_]^4−^ dimer (Ta2A part) in the crystal structure of **4**.

**Figure 5 molecules-28-04912-f005:**
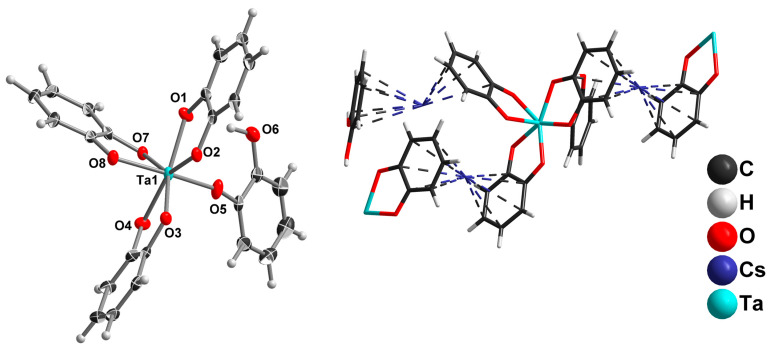
The structure of the [Ta(cat)_3_(Hcat)]^2−^ complex (ellipsoid plot with 50% probability) in the crystal structure of **5** (**left**); location of Cs^+^ cation in the crystal structure of **5** (**right**).

**Figure 6 molecules-28-04912-f006:**
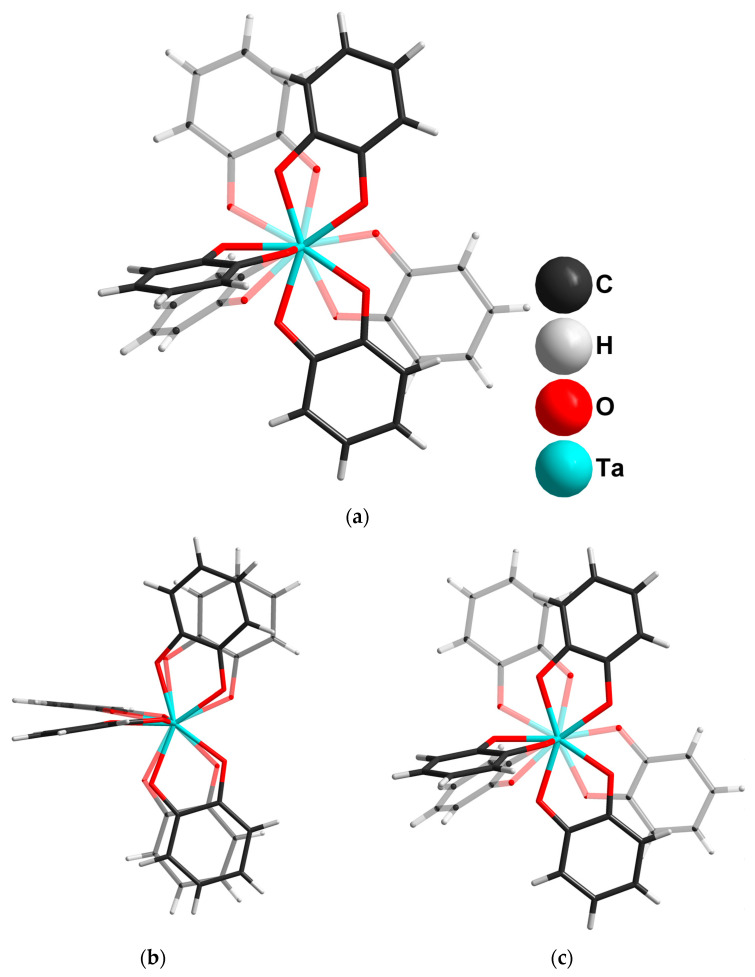
The comparison of dimers conformations in (**a**) **4**; (**b**) A-type; and (**c**) B-type in **6**.

**Figure 7 molecules-28-04912-f007:**
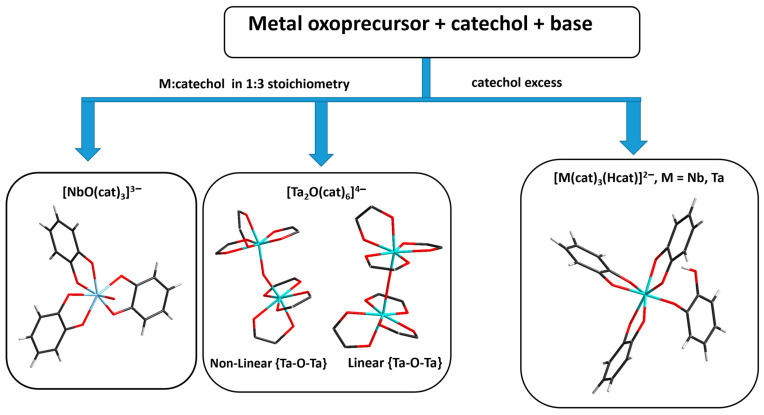
General scheme of the synthesis and structures of the Nb(V) and Ta(V) catecholate complexes.

## Data Availability

The data presented in this study are available in Appendix A. CCDC 2266543-2266548 contains the supplementary crystallographic data for **1**–**6**. These data can be obtained free of charge via https://www.ccdc.cam.ac.uk/structures/? or from the Cambridge Crystallographic Data Centre, 12 Union Road, Cambridge CB2 1EZ, UK; Fax: (+44) 1223-336-033.

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
