# Peer review of "Not So Similar: Different Ways of Nb(V) and Ta(V) Catecholate Complexation"

_molecules, 2023, doi:10.3390/molecules28134912_

Round 1
Reviewer 1 Report
The paper of Pavel A. Abramov and Maxim N. Sokolov is an interesting fundamental work on synthesis 6 new niobium and tantalum complexes with catechol. Authors succeeded growing crystals amenable for SC X-RAY analysis for all new compounds, determined their crystal structure. Complexes also were studied using XRPD, TGA, IR spectroscopy. Their structure and phase purity are not in doubt. However, there are a few questions and suggestions that can improve the text of the manuscript.
The article has a strange elemental analysis, moreover, it looks like a fake one. A few decades ago, this remark could be major and lead to an immediate rejection of the article.
For example, Line 205: Synthesis of K2(H2cat)2[Nb(cat)3(Hcat)]·H2O (2): С, H (%): 51.2, 3.9. Found С, H (%): 41.2, 5.2, 8.0. Here the authors apparently made a mistake and copied the data from synthesis 1. Since nitrogen, which the authors successfully discovered, is not contained in compound 2.
Line 218 and 225: Synthesis of Сs3[NbO(cat)3]·H2O (3): С, H, N (%): 31.2, 6.1, 12.4. Found С, H, N (%): 31.5, 6.1, 12.8. In a compound 3 that should not contain nitrogen atoms, the authors found nitrogen, and even within the statistically acceptable error. This again can be attributed to an error, since the authors calculate the content of atoms in the compound C20H46Cl4N7NbO9, but no such compound was obtained at all in this manuscript.
At present, elemental analysis is more or less suitable for organic substances, while measurement errors for clusters, coordination compounds, and organometallics are much higher than 0.4%. The authors of this manuscript have proved the structure by X-ray diffraction analysis, phase purity by X-ray powder diffraction analysis. As far as I know, Molecules does not strictly require the presence of elemental analysis. I advise dear authors, if there is no elemental analysis, do not write it in the experimental part in order to avoid such major mistakes.
I also have a few recommendations.
The title of the article contains the words "patterns", which is clearly associated with the phrase "powder diffraction patterns”. At first, I thought that the article would be about absolutely identically (isostructural) compounds of tantalum and niobium, which have different powder patterns. However, the word "patterns" means in this case "a different way of reaction". I am not a native English speaker and may be mistaken, however, the authors should check how accurately the title reflects the content of the article.
The paragraph on the redox activity of catechol (lines 45-48) looks unfinished and weak. Searching for relevant articles leads to excellent reviews and original articles on compounds of tantalum and niobium with redox active ligands, including various catechols. I give a few examples (I am not related to the authors of these articles and am not the author): https://doi.org/10.1039/C2DT32063K
According to the TGA data, compound 1 contains 4.4 water molecules, while according to X-ray diffraction data, there should be 4 water molecules. How can the authors explain the difference of 10% in this case and in other compounds?
Supplementary materials also contain IR spectra of the compounds. However, the authors do not talk about them in the article. However, IR spectra of redox-active ligands (catechol) are an accurate tool for determining the catecholate or semiquinone form.
What redox transitions did the authors expect for niobium(V) and the redox-active ligand in the catecholate form?
Line 42: Nb and ta
Line 148: via by
Author Response
The paper of Pavel A. Abramov and Maxim N. Sokolov is an interesting fundamental work on synthesis 6 new niobium and tantalum complexes with catechol. Authors succeeded growing crystals amenable for SC X-RAY analysis for all new compounds, determined their crystal structure. Complexes also were studied using XRPD, TGA, IR spectroscopy. Their structure and phase purity are not in doubt. However, there are a few questions and suggestions that can improve the text of the manuscript.
The article has a strange elemental analysis, moreover, it looks like a fake one. A few decades ago, this remark could be major and lead to an immediate rejection of the article.
For example, Line 205: Synthesis of K2(H2cat)2[Nb(cat)3(Hcat)]·H2O (2): С, H (%): 51.2, 3.9. Found С, H (%): 41.2, 5.2, 8.0. Here the authors apparently made a mistake and copied the data from synthesis 1. Since nitrogen, which the authors successfully discovered, is not contained in compound 2.
Line 218 and 225: Synthesis of Сs3[NbO(cat)3]·H2O (3): С, H, N (%): 31.2, 6.1, 12.4. Found С, H, N (%): 31.5, 6.1, 12.8. In a compound 3 that should not contain nitrogen atoms, the authors found nitrogen, and even within the statistically acceptable error. This again can be attributed to an error, since the authors calculate the content of atoms in the compound C20H46Cl4N7NbO9, but no such compound was obtained at all in this manuscript.
At present, elemental analysis is more or less suitable for organic substances, while measurement errors for clusters, coordination compounds, and organometallics are much higher than 0.4%. The authors of this manuscript have proved the structure by X-ray diffraction analysis, phase purity by X-ray powder diffraction analysis. As far as I know, Molecules does not strictly require the presence of elemental analysis. I advise dear authors, if there is no elemental analysis, do not write it in the experimental part in order to avoid such major mistakes.
Thank you very much for a careful check of the experimental details. These data were collected from students reports prepared 3 years ago. I didn’t check it carefully and this was my fault. I removed the strange data for K2(H2cat)2[Nb(cat)3(Hcat)]·H2O and Сs3[NbO(cat)3]·H2O.
I also have a few recommendations.
The title of the article contains the words "patterns", which is clearly associated with the phrase "powder diffraction patterns”. At first, I thought that the article would be about absolutely identically (isostructural) compounds of tantalum and niobium, which have different powder patterns. However, the word "patterns" means in this case "a different way of reaction". I am not a native English speaker and may be mistaken, however, the authors should check how accurately the title reflects the content of the article.
Agreed! We change “patterns” for “ways”.
The paragraph on the redox activity of catechol (lines 45-48) looks unfinished and weak. Searching for relevant articles leads to excellent reviews and original articles on compounds of tantalum and niobium with redox active ligands, including various catechols. I give a few examples (I am not related to the authors of these articles and am not the author): https://doi.org/10.1039/C2DT32063K
These data have been added to the Introduction part. Moreover, the Intro part has been updated.
According to the TGA data, compound 1 contains 4.4 water molecules, while according to X-ray diffraction data, there should be 4 water molecules. How can the authors explain the difference of 10% in this case and in other compounds?
SCXRD data give only 2 solvate water molecules. TGA gives c.a. 14% weight loss which is 4.4 H2O molecules. The use of standard Rules for Rounding gives 4 H2O. The same rules were applied for all formulas. This mathematical assumption covers sample preparation statistical errors appeared from different factors.
Supplementary materials also contain IR spectra of the compounds. However, the authors do not talk about them in the article. However, IR spectra of redox-active ligands (catechol) are an accurate tool for determining the catecholate or semiquinone form.
Typically, semiquinolate forms are unstable in air. Moreover, they have strong coloration like deep-green, deep-blue or deep-purple. In our case all compounds are colorless or light-yellow in color. These two parameters exclude the presence of any unusual catecholate redox forms. Some IR data have been highlighted in the main text to prove the redox state of the ligands.
What redox transitions did the authors expect for niobium(V) and the redox-active ligand in the catecholate form?
CV data will be a subject for further research.
Line 42: Nb and ta
Thank you very much! This was corrected!
Line 148: via by
Thank you very much! This was corrected!
Pavel A. Abramov
Novosibirsk, 19/06/23

Reviewer 2 Report
This is a straightfoward report on Nb(V) and Ta(V) complexes with catecholate ligands. Six new complexes are presented. The compounds were characterized by X-ray crystallography, powder X-ray diffraction, elemental analysis, IR spectroscopy and thermal analysis. I think the manuscript deserves publication in Molecules after some improvements have been made.
X-ray crystallography:
First of all, consider updating your SHELXL version. The latest version is SHELXL-2019/3. I honestly do not see a justification to exclude large numbers of reflections by using OMIT. The structures refine well against all data. OMIT 0 is even worse and should be removed in the final refinements. Setting the F2 values for all reflections with F2 < 0 zero may introduce a bias. The atom lists should be sorted in the SHELXL INS file: metal atoms first, followed by C, N and O. Then, the bonds will appear in the CIF as a chemist would write them (e.g. Nb1-O1 rather than O1-Nb1).
A chemical diagram or even an reaction scheme should be depicted for each compound.
After re-finalisation of the structure refinements (as indicated above), displacement ellipsoid plots (50 % probabillity level) should be shown for each structure (not just capped sticks). Those massive legends showing the colour of each atom type should be omitted. Suffice it to label metal and hetero atoms in the displacement ellipsoid plots.
Line 24: "non-non-organometallic" should probably be "non-organometallic"?
Line 42: "ta" should be "Ta"?
Line 53: "alkalis" should just be "alkali".
Line 71: I am not sure if there is evidence for positional disorder of NH4+ ions and H2O molecules in the crystal structure of 1. I would say it is just hard to distinguish both molecules based on the present data, because the electron densities are so similar.
Caption to Figure 2: I think the use the braces (curly brackets) is not in line with IUPAC nomenclature guidelines: "In formulae, square brackets, parentheses and braces are used in the following nesting order: [], [( )], [{( )}], [({( )})], [{({( )})}], etc".
Line 98: "CCDC search" should be "search of the Cambridge Structural Database" and the appropriate citation should be inserted.
Line 127: Maybe one should say that the distances vary within certain "range" (rather than interval).
Line 274: "SHELX" should be "SHELXL".
Line 297: Author contributions: I reckon the CRediT author statement "software" is not applicable here and hence should be removed. Obviously, no software was developed in this project.
Author Response
This is a straightfoward report on Nb(V) and Ta(V) complexes with catecholate ligands. Six new complexes are presented. The compounds were characterized by X-ray crystallography, powder X-ray diffraction, elemental analysis, IR spectroscopy and thermal analysis. I think the manuscript deserves publication in Molecules after some improvements have been made.
X-ray crystallography:
First of all, consider updating your SHELXL version. The latest version is SHELXL-2019/3. I honestly do not see a justification to exclude large numbers of reflections by using OMIT. The structures refine well against all data. OMIT 0 is even worse and should be removed in the final refinements. Setting the F2 values for all reflections with F2 < 0 zero may introduce a bias. The atom lists should be sorted in the SHELXL INS file: metal atoms first, followed by C, N and O. Then, the bonds will appear in the CIF as a chemist would write them (e.g. Nb1-O1 rather than O1-Nb1).
The difference between SHELXL-2019/3 (this was recently download from the official websource) and SHELXL-2017/1 is only in interaction of SHELXL libraries and Intel processor hyperthreading architecture. If you have AMD processor this is nothing… According to the comments I refined all structures in SHELXL-2019/2. The corresponding data were re-deposited to CCDC.
Excluding of reflections was done based on Fo-Fc data.
OMIT 0 helps with treating of reflections with negative intensities which are not presented in .LST file. If there are serious problems inside the refined data PLATON typically will indicate this meaning the use of OMIT “concrete reflections” instead of OMIT 0. If the PLATON check is clear this can be used.
The new sort rules were applied for all structures.
A chemical diagram or even an reaction scheme should be depicted for each compound.
The general scheme was added into the main text.
After re-finalisation of the structure refinements (as indicated above), displacement ellipsoid plots (50 % probabillity level) should be shown for each structure (not just capped sticks). Those massive legends showing the colour of each atom type should be omitted. Suffice it to label metal and hetero atoms in the displacement ellipsoid plots.
The ellipsoid model is not so popular now to illustrate the structures of presented compounds. If you open JACS or Angewandte Chemie you will not find any pictures in this model. The use of ellipsoid plots means a special attention to the concrete structural feature which is under discussion. PLATON check typically indicates the problems with ellipsoids and helps with their treatment. In other cases, Mercury is a powerful tool to illustrate all the data. We placed the corresponding ellipsoid plots to the main text and into the SI.
Line 24: "non-non-organometallic" should probably be "non-organometallic"?
Thank you very much! This was corrected!
Line 42: "ta" should be "Ta"?
Thank you very much! This was corrected!
Line 53: "alkalis" should just be "alkali".
In this case we mean "alkalis" as plural noun because we used KOH, CsOH and so on.
Line 71: I am not sure if there is evidence for positional disorder of NH4+ ions and H2O molecules in the crystal structure of 1. I would say it is just hard to distinguish both molecules based on the present data, because the electron densities are so similar.
The quality of X-ray diffraction data for 1 is not at the top level due to poor diffraction character of the crystals. I agreed that “hard to distinguish both molecules based on the present data” is more suitable in this situation. This was corrected!
Caption to Figure 2: I think the use the braces (curly brackets) is not in line with IUPAC nomenclature guidelines: "In formulae, square brackets, parentheses and braces are used in the following nesting order: [], [( )], [{( )}], [({( )})], [{({( )})}], etc".
Thank you very much! We lost “associate” at the end of this sentence. In the case of association, the use of curly brackets is common.
Line 98: "CCDC search" should be "search of the Cambridge Structural Database" and the appropriate citation should be inserted.
Thank you very much! This was corrected! The corresponding reference was added!
Line 127: Maybe one should say that the distances vary within certain "range" (rather than interval).
Ok!
Line 274: "SHELX" should be "SHELXL".
Thank you very much! This was corrected!
Line 297: Author contributions: I reckon the CRediT author statement "software" is not applicable here and hence should be removed. Obviously, no software was developed in this project.
Thank you very much! This was removed!
Pavel A. Abramov
Novosibirsk, 19/06/23
